# Nutritional Status Associated with Molecular Biomarkers, Physiological Indices, and Clinical Severity in Parkinson’s Disease Patients

**DOI:** 10.3390/ijerph17165727

**Published:** 2020-08-07

**Authors:** Tsu-Kung Lin, Yung-Yee Chang, Nai-Ching Chen, Chia-Wei Liou, Min-Yu Lan, Ying-Fa Chen, Chia-Liang Tsai

**Affiliations:** 1Department of Neurology, Kaohsiung Chang Gung Memorial Hospital, Chang Gung University College of Medicine, Kaohsiung 833, Taiwan; tklin@cgmh.org.tw (T.-K.L.); changyy7@gmail.com (Y.-Y.C.); m0124@cgmh.org.tw (N.-C.C.); cwliou@ms22.hinet.net (C.-W.L.); myl@ksts.seed.net.tw (M.-Y.L.); alphac@cgmh.org.tw (Y.-F.C.); 2Center for Parkinson’s Disease, Kaohsiung Chang Gung Memorial Hospital, Kaohsiung 833, Taiwan; 3Center for Mitochondrial Research and Medicine, Kaohsiung Chang Gung Memorial Hospital, Kaohsiung 833, Taiwan; 4Institute of Physical Education, Health and Leisure Studies, National Cheng Kung University, Tainan 701, Taiwan

**Keywords:** Parkinson’s disease, cholesterol, HbA1c, Unified Parkinson’s Disease Rating Scale, mini nutritional assessment, malnutrition

## Abstract

This study is intended to explore the associations between nutritional status and molecular biomarkers and the clinical severity of Parkinson’s disease (PD), as well as to examine the differences in related factors between PD patients with normal nutrition and those with at risk for malnutrition. A cross-sectional assessment of 82 consecutive outpatients with PD was conducted using the mini nutritional assessment (MNA), Unified Parkinson’s Disease Rating Scale (UPDRS), and the Hoehn and Yahr scale to determine the nutritional status, the clinical severity of PD, and the stage of the disease. Recordings of blood samples collected after 12 h of overnight fasting were also assessed in terms of serum levels of glycated hemoglobin (HbA1c), blood urea nitrogen (BUN), creatinine, cholesterol, high-density lipoprotein (HDL), low-density lipoprotein (LDL), hemoglobin (Hgb), folate, and vitamin B12. All participants were divided into normal nutrition and malnutrition risk groups via the MNA scores to compare the above-mentioned parameters. The results showed that the total MNA score was significantly correlated with some parts of the UPDRS scale (e.g., Sections 1 and 2) and the levels of HbAlc in PD patients and those with risk for malnutrition, with significantly lower weight and body mass index (BMI), and with lower levels of Hgb and HDL. Higher levels of cholesterol were observed in the malnutrition risk group as compared with the normal nutrition group. The findings suggest that the clinical severity of PD is associated with nutritional status. Body weight, BMI, and the levels of Hgb, cholesterol, and HDL could be, at least partially, important biological markers to monitor malnutrition and the progression of the disease.

## 1. Introduction

Parkinson’s disease (PD) is a chronic neurodegenerative disorder characterized by progressive degeneration of dopaminergic neurons located in the mesencephalon, resulting in loss of dopamine transporters in the striatum [1,2]. The cardinal symptoms and signs include bradykinesia, postural instability, shuffling gait, resting tremors, muscle rigidity, and curved posture [3]. The etiology and pathogenesis of PD are very complex and remain unclear at present. During the course of the disease, changes in nutritional status are suggested to be important potential factors associated with motor and non-motor complications (e.g., pneumonia and cachexia) induced by affecting the patient’s immune responses or long-term dopaminergic treatment [4,5,6,7,8]. A number of features of PD, such as progressive impairment of motor function (e.g., dysphagia) and non-motor symptoms (e.g., gastrointestinal dysfunctions), make normal dietary intake and digestion difficult, thus increasing the risks of undernutrition [9,10], weight loss, and an abnormally low body mass index (BMI) [11]. Therefore, there is a close relationship between nutritional status and PD [4,12], so nutritional assessments should be performed routinely [13].

Biochemical changes are associated with PD [14]. To understand the role of nutritional status on the motor- and non-motor clinical symptoms of PD, some molecular biomarkers were included in the present study. Cardiovascular risk factors (e.g., diabetes mellitus type II (DM2), and high cholesterol) appear to increase the risk of developing PD and contribute to more severe motor and non-motor features of PD [15,16]. Indeed, predating the introduction of levodopa therapy, the relationship between DM2 and PD was reported for several decades [17]. Peripheral insulin resistance correlates with the body mass index (BMI) and is prevalent in non-diabetic individuals with PD [18]. Associated biomarkers related to renal function and cholesterol-related dietary supplements (e.g., glycated hemoglobin (HbA1c), blood urea nitrogen (BUN), creatinine, cholesterol, high-density lipoprotein (HDL), high-density lipoprotein (LDL)) were thus examined in the PD patients with different nutritional statuses in the present study. In addition, anemia could be a risk factor for Parkinson’s disease [19]. The levels of hemoglobin (Hgb), vitamin B12, and folate are associated with anemia [14]. Hgb levels are reported to be related to the severity of PD [14]. Higher dietary intake of vitamin B12 and folate may inhibit the degeneration of neurons and lower the risk of PD through decreasing plasma homocysteine, in turn slowing neurotoxic processes via excitotoxicity and oxidative stress [20,21,22]. Accordingly, taking into account all of this information, the nutritional status of PD patients could be made more accurate through these biological markers because they can be affected by diet.

The prevention and treatment of nutritional deficiencies (e.g., micronutrients or vitamins) is an important issue in people with PD. Moreover, PD is a complex disease at the genetic, clinical, and molecular levels [23]. However, thus far, there is a lack of research specifically for patients with PD on the associations between nutritional status and the clinical severity of PD and molecular biomarkers. To clarify these aspects of the disease, we conducted a cross-sectional study examining the associations between the scores on the mini nutritional assessment (MNA) and the Unified Parkinson’s Disease Rating Scale (UPDRS) as well as biochemical markers, and compared the differences in these associated factors among PD patients with normal nutrition and those at risk for malnutrition.

## 2. Methods

### 2.1. Participants

A total of 82 consecutive outpatients diagnosed with idiopathic PD who fulfilled the modified Queen Square Brain Bank criteria for Parkinson’s disease were enrolled from the Center for Parkinson’s Disease at Chang Gung Memorial Hospital in Kaohsiung City. All of the PD patients were diagnosed clinically by at least one neurologist with an interest in movement disorders in the Neurology Department of Chang Gung Memorial Hospital. Patients with secondary parkinsonism or atypical clinical features, such as dementia at an early stage of the disease, pyramidal signs, cerebellar signs, gaze paresis, or significant focal lesions on computerized tomography or magnetic resonance imaging scans, were excluded. Patients with confirmed structural digestive-system diseases or other consumptive diseases such as infections and tumors were also excluded from this nutritional assessment. Each participant was informed of the purpose of the investigation, and the completion of the questionnaires was voluntary. All participants gave written informed consent, which was approved by the Institutional Review Board of Chang Gung Memorial Hospital (IRB No.: 201601182B0 & 201901361A3C501), Taiwan.

### 2.2. Study Design

All PD participants were asked to arrive at the Parkinson’s Disease center between 08:30 and 09:30 after avoiding food intake and smoking for at least 12 h and refraining from exercise for 24 h. After the participants underwent a blood draw under basal conditions, various questionnaires were filled out to collect data, including socio-demographic characteristics, clinical symptoms, and disease severity through the Unified Parkinson’s Disease Rating Scale (UPDRS) and the modified Hoehn and Yahr (H&Y) Stage questionnaires, and nutritional status was determined using the mini nutritional assessment (MNA) scale. Disease duration and levodopa equivalent doses were calculated based on medical records and a history review. Medication comorbidities of the enrolled patients, including DM2, hypertension, cerebrovascular diseases, cardiac diseases, gastric diseases, chronic renal diseases, hyperlipidemia, chronic obstructive pulmonary disease, anemia, gout, and cancers, were obtained by the research assistant. All the questionnaire assessments were performed by a research assistant who had undergone special training prior to the investigation, and face-to-face interviews were conducted with the patients and their spouses or children living with them.

### 2.3. Questionnaires and Scales

#### 2.3.1. Mini Nutritional Assessment (MNA)

The MNA is a rapid, comprehensive geriatric nutrition-assessment tool designed to evaluate early risk of malnutrition, with specificity, sensitivity, and positive predictive values being 98%, 96%, and 97%, respectively, according to the clinical status [24]. It serves as an extension of a standard evaluation of elderly patients in order to identify the risk of malnutrition before the manifestation of severe changes in weight, BMI, or albumin levels. The full format of the MNA scale consists of 18 items grouped into two sub-sections: six screening questions (Section 1) and 12 assessment questions (Section 2) [25]. The maximum score on the MNA assessment is 30 points, where 14 points are obtained through assessment questions, and 16 points can possibly be obtained by answering the screening questions. A total score of 24 points or more represent ‘‘normal nutritional status”, scores equal to or less than 23.5 were considered to be indicative of “risk of malnutrition”, and less than 17 points was categorized as a case of “malnutrition” [25]. As the MNA questionnaire includes simple measurements and brief questions, it can be carried out easily in approximately 10–15 min. This instrument has been demonstrated to be valid and reliable for nutritional status assessment of PD patients [4,26].

#### 2.3.2. Unified Parkinson’s Disease Rating Scale (UPDRS)

The UPDRS is the most commonly used scale to assess patients with PD. It is used to evaluate the clinical severity of PD [27]. It comprises four sections, including Section 1: non-motor symptoms (e.g., cognition, behavior, and mood status) (total 16 points), Section 2: activities of daily living (total 52 points), Section 3: motor examination (total 92 points), and Section 4: complications of therapy (total 23 points). The UPDRS is scored from a total of 147 points for the scale, where higher scores reflect worsening symptoms.

#### 2.3.3. Modified Hoehn and Yahr Stage

The modified Hoehn and Yahr (H&Y) stage is a widely used clinical rating scale to stage the broad categories of motor function in PD, with an increased stage representing disease progress. It contains five stages, where stage 0 indicates no visible symptoms of disease, and stage 5 indicates the most advanced symptoms of the disease, such as the inability to walk or being bedridden. Therefore, a higher stage indicates a greater level of functional disability [28].

### 2.4. Blood Sampling and Analysis

Fasting blood samples were obtained in vacutainer tubes after 12 h overnight fasting. Blood samples were collected in ethylenediamine tetraacetic acid (EDTA) tubes for further performing the HbA1c and hematological assay. HbA1c levels were measured using the affinity chromatography Premier Hb9210 analyzer (Trinity Biotech, Bray, Co.Wicklow, Ireland). The hematological analysis, including blood counts, reticulocytes, white blood cell (WBC) counts, WBC estimates, and leukocyte differentials, was conducted using the automated XN-9000 hematology analyzer (Sysmex, Kakogawa, Japan). Measurements of biochemical analytes were performed on the HITACHI LABOSPECT 008 system (Hitachi High-Tech Corporation, Hitachinaka-shi, Japan) using serum specimens. Biochemical analyses included cholesterol, HDL, LDL, BUN, creatinine, folate and vitamin B12 (Vit B12). All tests were performed in the Department of Laboratory Medicine at Kaohsiung Chang Gung Memorial Hospital, a College of American Pathologists (CAP) certified laboratory located in Taiwan.

### 2.5. Statistical Analysis

The results are reported as arithmetic mean ± SD. Any statistically significant correlations between permanent variables were assessed by means of a Spearman’s correlation test. The normal-nutrition and malnutrition-risk groups were compared using the χ^2^ test for categorical variables and an independent samples *t*-test for the mean value of the quantitative variables (e.g., age, height, weight, BMI, MNA scores, biochemical markers, and UPDRS scores). Statistically significant differences or correlations in all analytical procedures were set at *p* < 0.05 with a two-tailed approach.

## 3. Results

### 3.1. Demographic Data

Of the 82 cases of PD who underwent the nutritional assessment, the mean MNA score was 24.78 ± 2.27 (range 17.0–29.5). According to the MNA test, none of the patient population was suffering from malnutrition (MNS score < 17), but 29.27% of the patients (*n* = 24) were at risk for malnutrition (17 ≤ MNS score ≤ 23.5). However, nutritional status was satisfactory in 70.73% of the patients (*n* = 58). Therefore, all participants were classified into the normal-nutrition and malnutrition-risk groups.

As seen in Table 1, all PD patients were assessed with respect to the presence of other medical comorbidities, including hypertension, cardiac diseases, type 2 diabetes, anemia, and gastric diseases. Of all of the PD patients, anemia was present in seven patients (8.53%), hypertension was present in twenty-three patients (28.0%), cardiac diseases were present in seven patients (8.5%), type 2 diabetes mellitus was present in fourteen patients (17.1%), and gastric disease was reported in four patients (4.9%). In the malnutrition-risk group, anemia was present in five patients (20.83%), hypertension was present in five patients (20.8%), cardiac diseases were present in one patient (4.2%), diabetes mellitus was present in three patients (12.5%), and gastric disease was present in three patients (12.5%). Chi-square analyses failed to reveal any significant distribution differences in hypertention, cardiac diseases, type 2 diabetes, anemia, and modified Hoehn and Yahr (H&Y) stage between the normal-nutrition and malnutrition-risk groups. Only gastric diseases showed a significant between-group difference, with higher numbers being in the malnutrition-risk group.

### 3.2. Correlations

As seen in Appendix A, the total MNA score was significantly correlated with the scores of the UPDRS Sections 1 (*r* = −0.36, *p* = 0.001) and 2 (*r* = −0.35, *p* = 0.001), as well as with the total score on the UPDRS scale (*r* = −0.29; *p* = 0.009) in the PD patients. The levels of Hgb (*r* = 0.25, *p* = 0.022) and HbAlc (*r* = 0.23, *p* = 0.041) were also significantly correlated with the total MNA score. There were significant correlations between the UPDRS Section 1 and HDL (*r* = −0.29, *p* = 0.009) and Vit B12 (*r* = 0.24, *p* = 0.033) levels, and BUN levels and the UPDRS Section 3 (*r* = 0.28, *p* = 0.013) and total (*r* = 0.24, *p* = 0.030) scores.

In terms of the malnutrition-risk group, the total MNA score was significantly correlated with the scores on the UPDRS Section 1 (*r* = −0.47, *p* = 0.020) and 2 (*r* = −0.47, *p* = 0.019) and approached significance for the total score (*r* = −0.39, *p* = 0.058) of the UPDRS scale. Only the level of HbAlc was significantly correlated with the total MNA score (*r* = 0.62, *p* = 0.002), Sections 1 and 2, and the total scores on the UPDRS scale (all *r* = −0.51, *p* = 0.016).

### 3.3. The Normal-Nutrition Group versus the Malnutrition-Risk Group

As shown in Table 2, significant differences in weight and BMI were observed between the normal-nutrition and malnutrition-risk groups, with lower weight and BMI in the malnutrition-risk group relative to the normal-nutrition group. Significantly lower levels of Hgb and higher levels of cholesterol and HDL were observed in the malnutrition-risk group as compared with the normal-nutrition group. The levels of LDL in the between-group comparison only approached significance.

According to the results of UPDRS assessment, the complaints in the problems of activities of daily living (Section 2) were more significant in the malnutrition-risk group than in the normal-nutrition group. Additionally, the scores for the UPDRS Section 1 in the between-group comparison approached significance, indicating that the malnutrition-risk group showed poorer cognitive skills, behavior, and mood status as compared with the normal-nutrition group. However, there were no significant between-group differences on the motor examination (Section 3) and the complications related to therapy (Section 4).

## 4. Discussion

The present study was aimed toward investigating the associations between the nutritional status and molecular biomarkers and the clinical severity of PD, as well as to examine the differences in the factors associated with the normal-nutrition and malnutrition-risk groups. We found that the total MNA score was significantly correlated with some of the UPDRS scale items (e.g., Sections 1 and 2) as well as the level of HbAlc in PD patients and those with malnutrition-risk. In addition, the malnutrition-risk group relative to the normal-nutrition group showed significantly lower weight, BMI, and regular exercise habits, higher numbers of gastric comorbidity, lower levels of Hgb, and higher levels of cholesterol and HDL.

The side effects of long-term medication treatment for motor symptoms in PD may include poor appetite, nausea, dyskinesia, and sensory changes in the smell and taste of food [29,30]. In addition, non-motor symptoms (e.g., sleep disturbances, gastrointestinal-tract dysfunction, and psychiatric symptoms) shown in PD patients often result in negative psychological and physical impacts on the activities of daily living, leading to insufficient nutrient intake and exposing them to the risk of malnutrition [12,31,32,33]. In the present study, none of the PD patients were found to be malnourished, which bears similarity to Wang et al.’s [33] findings (1.71%), but the results were much lower than the results from other earlier reports [8,29,34]. However, although almost all of the participants were at H&Y stages 1 and 2 in this study, 29.27% of the PD patients were at risk for malnutrition, suggesting that PD can contribute to declining nutritional status [33]. This finding was much higher than that found in Barichella et al.’s [4] and Wang et al.’s [33] studies, in which the MNA assessment revealed that 22.9% and 19.66% of PD patients were at risk for malnutrition, respectively.

Previous studies have reported that a reduction in anthropometric parameters usually results from long-term undernutrition [8]. Nevertheless, in the current study, PD patients at risk for malnutrition still exhibited normal height, weight, and BMI, implying that anthropometric measurements may be insensitive in terms of detecting poor nutritional status. However, it is still worth noting that significantly lower weight and BMI were observed in the malnutrition-risk group as compared with the normal-nutrition group. Weight loss may lead to an increased risk of developing dyskinesias [35] that could, in turn, exacerbate the risk for malnutrition [30]. Weight loss in PD patients could be attributed to not only increased energy output owing to rigidity, tremors, and dyskinesia, but also reduced energy input [12]. At the same time, constipation has been reported to be associated with malnutrition and is one of the most important predictors [30,33]. Accordingly, nutritional balance designed to prevent weight loss and constipation should be considered in PD through nutritional counseling and interventions such as a Mediterranean-like diet and adequate dietary fiber and fluid intake [36,37].

According to the results of the UPDRS assessment, a statistically significant, but inverse correlation was determined between the MNA score and the UPDRS Sections 1 and 2 and total scores in PD patients and those at risk for malnutrition, suggesting that the clinical severity of PD, especially in terms of the non-motor symptoms (e.g., cognition, behavior, and mood status) and activities of daily living, may be affected by nutritional status. In addition, the complaints related to problems in activities of daily living (UPDRS Section 2) were more significant in the malnutrition-risk group than in the normal-nutrition group. The scores on the UPDRS Section 1 in the between-group comparison approached significance, indicating that the malnutrition-risk group showed poorer cognition, behavior, and mood status as compared with the normal-nutrition group. However, there were no significant between-group differences between the motor examination (UPDRS Section 3) and the complications related to therapy (UPDRS Section 4). The present findings partly concurred with Sheard et al.’s [30] findings that the scores of the UPDRS Sections 2 and 3 were significantly higher in the malnutrition group as compared with the good nutrition group. However, only UPDRS Sections 2 and 3 were measured in their study, and the nutritional status was assessed using the Patient-Generated Subjective Global Assessment tool. The present and previous findings may explain that there is a close relationship between the ability to perform daily tasks and the effects of nutritional deficits in PD patients.

Individuals with lower Hgb are more susceptible to PD [38]. According to the World Health Organization (WHO) criteria, Hgb levels <13.0 g/dL in men or <12.0 g/dL in women are diagnosed with anemia [39]. In total, seven PD patients had anemia based on the WHO criteria in the present study, including five cases in the malnutrition-risk group and two in the normal-nutrition group. Indeed, Deng et al. [14] found that lower Hgb levels were found in PD patients in comparison with controls. In addition, there was a statistically significant between-group difference with respect to this biomarker, with median Hgb levels being determined as lower in the malnutrition-risk group. Madenci et al. [40] found that, as the duration of the PD symptoms increased, serum Hgb and iron levels decreased. As lower Hgb may be a risk factor for PD [19] and is related to peripheral iron metabolism [14], which may be important in PD pathogenesis [41], further research seems to be warranted considering the iron deficiency anemia or lower iron absorption that accompanies malnutrition in advanced PD. In addition, although no statistically significant between-group differences with respect to the folate and vitamin B12 were found, there was a significant correlation between the UPDRS Section 1 score and vitamin B12 levels in PD patients. This result is partly in line with the Madenci et al.’s [40] finding of a significant negative correlation between H&Y scores and vitamin B12 levels. Therefore, it is worth noting that a decrease in serum vitamin B12 levels could accelerate PD progression secondarily by indirectly increasing the dopaminergic neuron degeneration owing to increases in homocysteine levels [13,40].

In the present study, both groups’ HDL levels were above threshold values. Nevertheless, the malnutrition-risk group showed abnormal levels of cholesterol (>200 mg/dL), and the levels of LDL were not within normal limits (<100 mg/dL) in either group, with the malnutrition-risk group relative to the normal-nutrition group exhibiting significantly higher LDL levels. Although previous studies investigating relatively small sample sizes reported that lower LDL levels are associated with a higher occurrence of PD [42] and that higher serum total cholesterol is associated with a decreased PD risk [20], in a large prospective study, an increased risk of PD associated with serum high total cholesterol levels was observed [43]. However, in the present study, the normal-nutrition and malnutrition-risk groups were almost statistically indifferent with respect to LDL levels, and significant between-group differences were found for mean cholbelong to the uncertainty sphere esterol and HDL. Hypercholesterolemia leads to other diseases (e.g., cardiovascular diseases) and further exacerbates PD symptoms. Additionally, given that the brain is the most cholesterol-rich organ in the entire body, and the fact that the aggregation of α-synuclein may play a critical role in the pathogenesis of PD [43], change in cholesterol composition using nutritional interventions (e.g., decreased intake of saturated and animal fat) or cholesterol-lowering agents (e.g., statins) seems to be an effective way to reduce the levels of α-synuclein and further delay the progression of PD.

While no statistically significant differences were observed between the normal-nutrition and malnutrition-risk groups with respect to the HbAlc, BUN, and creatinine levels, HbAlc was significantly correlated with the total MNA score in the PD patients, as well as with the total MNA score and the UPDRS Sections 1 and 2 and total scores in the malnutrition-risk group. Mollenhauer et al. [23] explored various baseline predictors of different modalities for disease progression in early PD after 24 and 48 months and found that HbAlc is a predictor of greater progression of cognitive decline in PD patients. However, the finding was not significant when those with diabetes were excluded. HbA1c is associated with serum urate [44], which is involved in the insulin resistance prevalent in PD [18,45]. Elevated serum urate levels are reported to predict worse motor and cognitive outcomes in PD [23]. A history of diabetes has been associated with an increased risk of PD [46]. Therefore, the progress of the disease may be slowed down in PD patients, especially in terms of cognitive functions, via early nutritional interventions aimed toward lowering HbA1c levels.

There are limitations to this approach that must be addressed. First, given that almost all of the PD patients fell within the H&Y 1 and 2 categories in the current study, the application of the present findings to PD patients in the other H&Y categories should be taken with caution, as such patients with late stage (i.e., H&Y 3 and 4 categories) may have a greater degree of under-nutrition, with deviant fat and muscle mass accounting for the change [8,47]. Second, increased energy expenditure or impaired absorption/insufficient intake from the gastrointestinal tract could be potential factors related to the poor nutritional status in PD [48]. A previous study found that vomiting and constipation were significantly more common in a poor-nutrition group as compared with a well-nourished group, and there was a strong correlation found between the two factors and malnutrition, suggesting that gastrointestinal dysfunction (e.g., vomiting and constipation) insidiously leads PD patients to the edge of malnutrition [33]. Indeed, higher numbers of gastric comorbidity were found in the malnutrition-risk group as compared with the normal-nutrition group in the present study. Along these lines, it would be informative in the future to understand the nutritional status of PD through not only a questionnaire, but also an examination of gastrointestinal function. Third, in the present study, lower numbers of regular exercise habits were observed in the malnutrition-risk group relative to the normal-nutrition group. However, regular physical activity could decrease the homocysteine levels [49], which are significantly associated with nutritional status in PD patients [50]. In addition, increasing levodopa (L-dopa) equivalent daily dosages were associated with an increasing probability of malnutritional risk in PD patients [51]. An avenue for future research is to examine the possibility of homocysteine/L-dopa levels and nutritional status interaction in the PD population. Fourth, motor function deterioration coupled with lean body mass loss might result in negative effects on nutritional status in PD [30]. Only body weight rather than lean body mass was measured in the current study. Moreover, there were no age- and sex-matched healthy controls in the present study to compare differences in nutritional status. Lastly, the degree of depression could be a latent threat to the nutritional status in PD patients [33]. Further investigations of the above-mentioned issues are warranted.

## 5. Conclusions

As with other neurodegenerative diseases, nutritional and neurologic status may interact and in turn affect each other in PD. Increased energy expenditure at rest is correlated with the degree of muscle rigidity in PD patients [8]. Barichella et al. [4] found that the proportion of PD patients at risk for malnutrition increased sharply in three years from 22.9 to 34.3%, possibly owing to a lack of the knowledge regarding PD-related nutritional problems among medical staff. Consequently, the diagnosis and treatment of PD should involve more attention on non-motor symptoms (e.g., nutritional status) to prevent these patients from progressing to malnutrition or becoming at risk for malnutrition.

## Figures and Tables

**Table 1 ijerph-17-05727-t001:** Summary of the characteristics of the total Parkinson’s disease (PD) patients, and PD patients with normal nutrition and malnutrition risk grouped using the MNA scale.

	Total (*n* = 82)	Normal-Nutrition (*n* = 58)	Malnutrition-Risk (*n* = 24)	*p*
Gender (male/female)	58/24	46/12	12/12	0.008 *
Age (years)	67.40 ± 9.06	67.74 ± 8.08	66.58 ± 11.23	0.602
Height (cm)	163.72 ± 9.21	164.57 ± 8.85	161.67 ± 9.92	0.196
Weight (kg)	64.23 ± 10.42	67.34 ± 9.65	56.71 ± 8.24	<0.001 *
BMI (kg/m^2^)	23.93 ± 3.10	24.82 ± 2.58	21.77 ± 3.24	<0.001 *
Education (years)	11.52 ± 4.78	11.98 ± 5.03	10.42 ± 3.98	0.179
Vegetarian	7	4	3	0.283
Smoking	3	2	1	0.875
Alcohol (social drinking)	12	11	1	0.085
Walking aids	5	3	2	0.796
Regular exercise	62	49	13	0.004 *
H&Y 1	7	4	3	0.242 ^#^
H&Y 2	71	53	18	
H&Y 3	2	1	1	
H&Y 4	1	0	1	
H&Y 5	1	0	1	
Hypertension	23	18	5	0.349
Cardiac diseases	7	6	1	0.362
Type 2 diabetes	14	11	3	0.479
Anemia	7	2	5	0.090
Gastric diseases	4	1	3	0.039 *

BMI: body mass index; H&Y: modified Hoehn and Yahr Stage; MNA: mini nutritional assessment; *p*-values denote the differences between the normal-nutrition and malnutrition-risk groups; * *p* < 0.05; ^#^ A distribution difference in overall five modified H&Y stages between the normal-nutrition and malnutrition-risk groups.

**Table 2 ijerph-17-05727-t002:** Analysis of the factors associated with the normal-nutrition and malnutrition-risk groups.

	Normal-Nutrition	Malnutrition-Risk	*p*
MNA scores			
Section 1 *	13.41 ± 0.80	11.54 ± 1.50	<0.001
Section 2 *	12.53 ± 0.99	10.42 ± 1.53	<0.001
Total *	25.95 ± 1.12	21.96 ± 1.81	<0.001
Cholesterol (mg/dL) *	184.20 ± 36.43	208.67 ± 39.27	0.009
HDL (mg/dL) *	60.88 ± 13.51	52.68 ± 11.94	0.008
LDL (mg/dL) ^#^	113.63 ± 32.42	127.63 ± 34.75	0.087
HbAlc (%)	5.75 ± 1.78	5.47 ± 1.90	0.537
BUN (mg/dL)	15.76 ± 6.03	15.52 ± 5.13	0.870
Creatinine (mg/dLl)	0.92 ± 0.19	0.88 ± 0.20	0.414
Hgb (gm/dL) *	14.33 ± 1.35	13.48 ± 1.36	0.011
Folate (ng/mL)	12.26 ± 7.25	12.33 ± 9.69	0.974
VitB12 (pg/mL)	685.64 ± 399.14	726.71 ± 788.56	0.764
UPDRS scores			
Section 1 ^#^	1.50 ± 1.29	2.13 ± 1.75	0.077
Section 2 *	5.86 ± 4.37	9.00 ± 5.90	0.009
Section 3	26.52 ± 7.77	29.38 ± 12.42	0.304
Section 4	1.02 ± 1.57	1.17 ± 2.14	0.727
Total	34.90 ± 11.81	41.67 ± 18.49	0.107

BMI: body mass index; MNA: mini nutritional assessment; Hgb: hemoglobin; HDL: high-density lipoprotein; LDL: low-density lipoprotein; HbA1c: glycated hemoglobin; BUN: blood urea nitrogen; UPDRS: Unified Parkinson’s Disease Rating Scale; * *p* < 0.05; ^#^ approached significance.

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
