# Peer review of "Nutritional Status Associated with Molecular Biomarkers, Physiological Indices, and Clinical Severity in Parkinson’s Disease Patients"

_ijerph, 2020, doi:10.3390/ijerph17165727_

Round 1

Reviewer 1 Report

The authors needs to remark the limitation of the papers at the end of Discussion section.

For example how the severity of the PD can affect this conclusions. The scope of the present work only is related to I-II stages of Hoehn & Yahr scale.

In my opinion, this paper is relevant in the research of risk factors for Parkinson Disease.

There is a lack of research about the associations between nutritional status and the clinical severity of PD and molecular biomarkers. They conducted a cross-sectional study examining the associations between the scores on the Mini Nutritional Assessment (MNA) and the Unified Parkinson’s Disease Rating Scale (UPDRS) as well as biochemical markers and compared the differences in these associates factors among PD patients with normal nutrition and those at risk for malnutrition.

Insufficient attention has been paid to nutritional and impaired gastrointestinal absorption of food and nutrients and its interaction with Parkinson progression.

The paper has solid evidences, reached logic conclusions and its publication can help to understand better the Parkinson Disease. But their results are limited to stages I and II of Hoehn & Yahr scale where the severity is less than the later stages. This is a limitation to be considered.

Reviewer 2 Report

The manuscript is well structured and presents interesting results. However, there are some aspects to improve, both in methodological and presentation matters.

The title and the abstract text should be re-considered. Only Hb or cholesterol are molecular biomarkers, the other factors (wheight-BMI), are physiological.

If the objective of the work is to relate three parameters (molecular biomarkers-nutritional status-severity of the disease), is necessary to carry out another type of statistical analysis that allow the analysis of the relationship among three variables, for example, analysis of principal components Other option is relates the three variables by twos.

The results showed in 3.2 (Correlation) should be presented as a table, and the data used to obtain them must be shown, or attached as supplementary material.

Table 1:

The MNA abreviation is on the foot, but not in the table. If the patients are grouped using MNA should be indicated in the table, preferably in the head.

The results obtained regarding alcohol consumption, hypertension, cardiovascular disease and diabetes should be discussed, since it gives the impression that they reduce the risk of malnutrition. Even should be determined if the results are significant.

Table 2:

What is the meaning of a-f / g-r sections? This information is not available in the manuscript

In these results only Hgb is related to the malnutrition risk, but not HbAlc, please discuss this result.

Discussion

Constipation is related only partially with malnutrition. Also the lack of dopamine, sedentarism or drinking problems could explain this symptom.

p8-l294. The reference appears complete in parentheses (Sheard et al., 2013) [30]

References

The number appears two time in the list of references eg. 33. [33] Wang, G,; ....

Round 2

Reviewer 2 Report

The quality of the manuscript is now appropriate for publication in its current form.
It is only necessary to review the English and the reference list format (duplicate numbers).